# Diagnostic Performance of [^18^F]F-FDG Positron Emission Tomography (PET) in Non-Ophthalmic Malignant Melanoma: A Systematic Review and Meta-Analysis of More Than 10,000 Melanoma Patients

**DOI:** 10.3390/cancers16010215

**Published:** 2024-01-02

**Authors:** Nazanin Zamani-Siahkali, Seyed Ali Mirshahvalad, Christian Pirich, Mohsen Beheshti

**Affiliations:** 1Division of Molecular Imaging and Theranostics, Department of Nuclear Medicine, University Hospital, Paracelsus Medical University, 5020 Salzburg, Austria; n-zamanisiahkali@razi.tums.ac.ir (N.Z.-S.); c.pirich@salk.at (C.P.); m.beheshti@salk.at (M.B.); 2Department of Nuclear Medicine, Shariati Hospital, Tehran University of Medical Sciences, Tehran 1461884513, Iran; 3Joint Department of Medical Imaging, University Medical Imaging Toronto, University Health Network, Sinai Health System, Women’s College Hospital, University of Toronto, Toronto, ON M5G 2N2, Canada

**Keywords:** FDG, PET, melanoma, staging, metastasis, sensitivity, specificity, systematic, meta-analysis, PRISMA

## Abstract

**Simple Summary:**

Malignant melanoma is a highly aggressive malignancy, accounting for the majority of skin cancer deaths. Its global incidence rate has been increasing rapidly over the past few decades, and accurate disease diagnosis is crucial for optimal patient management. Imaging modalities, such as positron emission tomography (PET), play a pivotal role in accomplishing this objective. Based on current guidelines, [^18^F]F-FDG PET is recommended for the assessment of distant metastases in advanced-stage melanoma patients. In addition, it is indicated in patient follow-up. However, to our knowledge, there is no comprehensive investigation in the literature reporting the real-world performance of [^18^F]F-FDG PET in melanoma patients across a broad spectrum of clinical scenarios. Thus, we aimed to describe the performance of [^18^F]F-FDG PET in a wide range of detection-based clinical scenarios (e.g., initial staging, re-staging, locoregional detection, distant metastasis diagnosis) by conducting a systematic review and meta-analysis of the existing literature.

**Abstract:**

We described the diagnostic performance of [^18^F]F-FDG-PET in malignant melanoma by conducting a comprehensive systematic review and meta-analysis of the existing literature. The study was designed following PRISMA-DTA. Original articles with adequate crude data for meta-analytic calculations that evaluated [^18^F]F-FDG-PET and compared it with a valid reference standard were considered eligible. The pooled measurements were calculated based on the data level (patient/lesion-based). Regarding sub-groups, diagnostic performances were calculated for local, regional and distant involvement. The bivariate model was employed to calculate sensitivity and specificity. The initial search resulted in 6678 studies. Finally, 100 entered the meta-analysis, containing 82 patient-based (10,403 patients) and 32 lesion-based (6188 lesions) datasets. At patient level, overall, [^18^F]F-FDG-PET had pooled sensitivity and specificity of 81% (95%CI: 73–87%) and 92% (95%CI: 90–94%), respectively. To detect regional lymph node metastasis, the pooled sensitivity and specificity were 56% (95%CI: 40–72%) and 97% (95%CI: 94–99%), respectively. To detect distant metastasis, they were 88% (95%CI: 81–93%) and 94% (95%CI: 91–96%), respectively. At lesion level, [^18^F]F-FDG-PET had a pooled sensitivity and specificity of 70% (95%CI: 57–80%) and 94% (95%CI: 88–97%), respectively. Thus, [^18^F]F-FDG-PET is a valuable diagnostic modality for melanoma assessment. It was accurate in various clinical scenarios. However, despite its high specificity, it showed low sensitivity in detecting regional lymph node metastasis and could not replace lymph node biopsy.

## 1. Introduction

Melanoma is a highly aggressive type of cancer originating from the uncontrolled growth and malignant transformation of melanocytes [1,2]. While comprising only about 5% of all skin malignancies, melanoma accounts for the majority of skin cancer deaths, a healthcare burden that cannot be overlooked [3]. Its global incidence rate has been increasing rapidly over the past few decades in both sexes, accounting for 324,635 cases globally based on the 2020 GLOBOCAN data [1,4]. Thus, with a continuous rise in the number of patients, accurate staging, treatment monitoring and early detection of recurrences are crucial for the optimal management of the disease. Imaging modalities, such as positron emission tomography (PET), play a pivotal role in accomplishing this objective.

Over the past two decades, PET, in combination with computed tomography (CT), has considerably improved our ability to evaluate cancer across various aspects, ranging from primary detection to response monitoring. ^18^F-fluorodeoxyglucose ([^18^F]F-FDG) is a glucose analogue serving as the most extensively employed tracer for PET imaging. [^18^F]F-FDG PET/CT is a non-invasive imaging modality that can provide both anatomic and functional information, allowing for improved accuracy in tumor localization and characterization. In addition, it is capable of depicting and tracking metabolic changes in whole-body acquisition [5].

Melanoma is categorized by the American Joint Committee on Cancer (AJCC; 8th edition) using the TNM staging system into localized disease (stage I and II), node-positive disease (stage III), and advanced or metastatic disease (stage IV) [6]. Based on the European consensus-based interdisciplinary guideline [6], [^18^F]F-FDG PET/CT is recommended for the assessment of distant metastases in patients with stage III and IV melanoma or poor-prognosis stage IIC patients. In addition, [^18^F]F-FDG PET/CT is indicated in the follow-up of stage IIC-IV melanoma patients [6].

However, to our knowledge, there is not a comprehensive investigation in the literature reporting the real-world performance of [^18^F]F-FDG PET in melanoma patients across a wide range of clinical scenarios. Having said that, while not focusing on [^18^F]F-FDG PET/CT in particular, a Cochrane Library review study investigated different imaging modalities’ performance in melanoma (limited to cutaneous melanoma) [7]. However, they only studied papers that allowed direct/indirect *comparisons* between imaging modalities, making their conclusion somehow limited regarding the real-world performance of [^18^F]F-FDG PET. Thus, in this study, we aimed to exclusively present the diagnostic performance of [^18^F]F-FDG PET in a wide range of detection-based clinical scenarios of malignant melanoma (e.g., initial staging, re-staging, locoregional detection, distant metastasis diagnosis) by conducting a comprehensive systematic review and meta-analysis of the existing literature. We excluded ophthalmic melanoma due to its different nature and behavior.

## 2. Materials and Methods

This study was designed following the Preferred Reporting Items for Systematic Review and Meta-Analysis of Diagnostic Test Accuracy studies (PRISMA-DTA) [8].

### 2.1. Search Strategy

A systematic search was conducted on the three main medical literature databases, Scopus, Web of Science, and PubMed, until 1 January 2023. The search was performed using database-specific Boolean search strategies, covering all key terms (both text and MeSH terms) to include the mainstem search strategy: [“melanoma AND (pet/ct OR pet/mr* OR positron OR *fdg*)”]. No time or language limit was considered. The electronic searches were augmented by searching through the included articles’ references to add any relevant study.

### 2.2. Study Selection

Published original articles that met the following inclusion criteria were considered eligible for inclusion in the meta-analysis:(1)Evaluated [^18^F]F-FDG PET, PET/CT, or PET/MRI as the diagnostic method to detect tumoral lesions in non-ophthalmic melanoma patients, including cutaneous melanoma, mucosal melanoma, and melanoma of unknown primary. For the sake of this review, we will refer to [^18^F]F-FDG PET, PET/CT, and PET/MRI with general terminology of *[^18^F]F-FDG PET* in the rest of the manuscript, unless particularly mentioned.(2)Compared [^18^F]F-FDG PET with histopathology, follow-up, or asynchronous multimodality imaging as the reference standard.(3)Provided adequate raw data to calculate true positive (TP), true negative (TN), false positive (FP), and false negative (FN) results.

The exclusion criteria included meeting abstracts, abstracts without full articles, and unpublished/not peer-reviewed articles. Duplicated studies were excluded. The titles and abstracts were reviewed to check the studies’ eligibility to enter the full-text review phase. Two reviewers (NZS and SAM with three and four years of expertise in molecular imaging, respectively) independently screened titles and abstracts separately to identify relevant citations. In case of any discrepancy, a third reviewer (MB; molecular imaging expert) decided on the eligibility. To assess the quality of the studies, the two earlier-mentioned reviewers used the Quality Assessment of Diagnostic Accuracy Studies-2 (QUADAS-2) tool to exclude the low-quality articles in consensus.

### 2.3. Data Extraction

For the remaining studies, after full-text review, data extraction was performed by two reviewers (NZS and SAM). In addition to the raw data (TP, TN, FP, FN) gathering, extracted data included the title, first author’s name, the aim of the study, utilized modality, reference standard, level of the analyses (patient-based vs. lesion-based), number of patients/lesions, status of the evaluated patients (initial staging versus secondary staging, including restaging, follow-up, surveillance, and recurrence), female/male numbers, location of the primary lesions, and stage of the patients.

### 2.4. Statistical Analysis

The retrieved data was presented as either a mean or a percentage, depending on whether the variables were continuous or categorical. The hierarchical method was employed to pool the random effect model’s diagnostic performance measures of sensitivity, specificity, positive likelihood ratio (LR), and negative LR from the derived two-by-two contingency tables. The pooled measurements were calculated based on the level of the data (patient-based versus lesion-based), and the general groups/subgroups were defined based on these levels. To note, if a study provided data for both levels of analysis, we extracted both of the datasets (thus, the number of analyzed datasets will be more than the number of included manuscripts). Regarding sub-group analysis, diagnostic performances were calculated for local, regional, and distant involvement. For regional lymph node involvement, we also limited the data to those evaluated with histopathology to report a more robust performance of [^18^F]F-FDG PET. If provided, we sub-grouped patients based on their status when evaluated (initial staging versus secondary staging) for each purpose.

The bivariate model was employed to determine the key findings for sensitivity and specificity, along with their corresponding 95% confidence intervals (CI), in order to account for any variations within and across studies. The summary receiver operating characteristic (SROC) model was utilized to construct the hierarchical SROC curve and determine the area under the curve (AUC). In order to demonstrate the degree of uncertainty surrounding the pooled sensitivity and specificity, we further constructed the 95% confidence region and prediction region. Additionally, scatterplots were created to analyze the overall results. The heterogeneity calculation involved the utilization of Higgins’ I^2^ statistics and Cochran’s Q test. The subsequent interpretation of the data adhered to the guidelines outlined by Cochrane [9]. The analyses were conducted utilizing the “Midas” and “Metaprop” modules inside the STATA 16 (StataCorp) software [10,11]. *p*-value < 0.05 was considered to indicate a statistically significant difference.

## 3. Results

### 3.1. Study Characteristics

The initial search resulted in 6678 studies. Of these, 3061 studies were entered in the title/abstract review after duplication removal, from which 2689 studies were excluded. During the full text review of the remaining 372 articles, 269 were considered ineligible based on the above-mentioned selection criteria. Finally, 100 studies [12,13,14,15,16,17,18,19,20,21,22,23,24,25,26,27,28,29,30,31,32,33,34,35,36,37,38,39,40,41,42,43,44,45,46,47,48,49,50,51,52,53,54,55,56,57,58,59,60,61,62,63,64,65,66,67,68,69,70,71,72,73,74,75,76,77,78,79,80,81,82,83,84,85,86,87,88,89,90,91,92,93,94,95,96,97,98,99,100,101,102,103,104,105,106,107,108,109,110] were found to be eligible to enter the meta-analytic calculations, including 82 patient-based datasets [12,13,14,15,16,17,18,19,20,21,22,23,24,25,26,27,28,33,35,36,37,39,40,41,42,43,44,46,48,49,50,52,53,54,56,57,58,59,60,61,62,63,64,65,66,67,69,70,71,72,73,74,75,76,77,78,81,82,83,84,85,86,88,89,90,91,93,94,95,97,99,100,101,102,103,107,108,109,110] and 32 lesion-based datasets [29,30,31,32,34,37,38,45,47,50,51,55,60,61,63,64,68,76,79,80,84,87,92,96,98,102,104,105,106]. In total, 10,403 patients and 6188 lesions entered the final population. Figure 1 illustrates the flow diagram for the inclusion process based on PRISMA [111]. Table 1 shows the detailed extracted data from each study.

### 3.2. Diagnostic Performances

#### 3.2.1. Patient-Level Diagnostic Performances

At the patient-level analysis, overall, [^18^F]F-FDG PET had a pooled sensitivity and specificity of 81% (95%CI: 73–87%) and 92% (95%CI: 90–94%), respectively. Positive and negative LRs were 10.1 (95%CI: 7.7–13.3) and 0.21 (95%CI: 0.15–0.29), respectively. In initial versus secondary staging, [^18^F]F-FDG PET had pooled sensitivities of 60% (95%CI: 43–75%) versus 90% (95%CI: 83–94%) and pooled specificities of 95% (95%CI: 91–97%) versus 89% (95%CI: 86–91%), respectively. Figure 2 and Figure 3 show the forest plots of the initial and secondary staging at the patient level, respectively. The SROC curves are provided in Figure 4.

In terms of detecting regional lymph node metastasis (both in initial and restaging settings), [^18^F]F-FDG PET revealed a pooled sensitivity and specificity of 56% (95%CI: 40–72%) and 97% (95%CI: 94–99%), respectively. Figure 5 shows the SROC curve and the LR scattergram.

When limiting data only to patients evaluated with the histopathology gold standard, these measures were 44% (95%CI: 27–63%) and 96% (95%CI: 90–99%), respectively. Considering only the detection of regional metastasis in patients referred for initial staging, [^18^F]F-FDG PET had a pooled sensitivity and specificity of 38% (95%CI: 22–56%) and 96% (95%CI: 90–99%), respectively. Again, when limiting data to histopathologically assessed patients, these measures were 37% (95%CI: 20–57%) and 96% (95%CI: 88–99%), respectively.

To detect patients with distant metastatic involvement, [^18^F]F-FDG PET showed a pooled sensitivity and specificity of 88% (95%CI: 81–93%) and 94% (95%CI: 91–96%), respectively. Figure 6 shows the SROC curve and LR scattergram. For initial versus secondary staging, these measures were 87% (95%CI: 69–95%) versus 91% (95%CI: 79–97%) and 96% (95%CI: 92–98%) versus 91% (95%CI: 79–96%), respectively.

#### 3.2.2. Lesion-Level Diagnostic Performances

In the lesion-level analysis (detecting all primary/recurrent malignant tumors, regional metastatic lymph nodes, and distant metastases), [^18^F]F-FDG PET had a pooled sensitivity and specificity of 70% (95%CI: 57–80%) and 94% (95%CI: 88–97%), respectively. Positive and negative LRs were 11.1 (95%CI: 5.8–21.5) and 0.32 (95%CI: 0.22–0.47), respectively. Drawn forest plots for the lesion-level pooled data are provided in Figure 7. At initial staging, the pooled sensitivity and specificity were 43% (95%CI: 17–74%) and 96% (95%CI: 79–99%), respectively.

In the regional lymph node metastatic lesions (detecting every metastatic regional lymph node) subgroup analysis, the pooled sensitivity and specificity were 38% (95%CI: 19–62%) and 98% (95%CI: 91–100%), respectively. Limiting data to initial staging, these measures were 18% (95%CI: 6–44%) and 99% (95%CI: 74–100%), respectively. Furthermore, when considering only lesions evaluated using the histopathology gold standard in the initial staging, sensitivity and specificity were 14% (95%CI: 6–27%) and 100% (95%CI: 70–100%), respectively.

To detect all distant metastatic lesions, [^18^F]F-FDG PET showed a pooled sensitivity and specificity of 84% (95%CI: 72–92%) and 93% (95%CI: 83–97%), respectively. Considering only patients referred for initial staging, these measures were 82% (95%CI: 69–90%) and 88% (95%CI: 57–98%), respectively.

## 4. Discussion

To the extent of our knowledge, this is the largest systematic review and meta-analysis to date that exclusively evaluates the diagnostic performance of [^18^F]F-FDG PET imaging in various clinical aspects of non-ophthalmic malignant melanoma. This meta-analysis included 100 studies with a substantial number of melanoma patients, totaling more than 10,000, which allowed for robust and reliable pooled results. We present the pooled diagnostic performance of [^18^F]F-FDG PET, which, when possible, was analyzed separately at the per-patient and per-lesion levels. The overall pooled sensitivity and specificity of [^18^F]F-FDG PET were calculated as 81% and 92% for patient-based analysis, respectively, and 70% and 94% for lesion-based analysis. Although these findings may indicate that [^18^F]F-FDG PET can be an accurate imaging tool for diagnostic purposes in malignant melanoma, it is important to interpret the *overall* pooled values cautiously in clinical practice. Hence, to derive more clinically relevant results, we categorized the findings into different subgroups based on their respective clinical settings and the reference standard.

When categorizing studies into initial versus secondary (restaging, follow-up, surveillance, recurrence) staging groups, [^18^F]F-FDG PET exhibited significantly higher sensitivity (90% vs. 60%) and comparable specificity (90% vs. 95%) in the secondary staging group. As noted by several studies [21,90], it appears that [^18^F]F-FDG PET may not be as valuable in earlier stages of melanoma as it is in more advanced stages, possibly due to the smaller size of lesions and a lower rate of metastasis. This could explain our calculated lower performance of [^18^F]F-FDG PET in initial versus secondary staging to some extent, where almost all melanoma patients included in the latter had advanced disease, making the visualization of larger regional lymph nodes and distant metastasis more probable.

Regarding regional metastasis detection, [^18^F]F-FDG PET showed low sensitivity, especially on a per-lesion basis (38%), though having satisfactorily high specificity (96%). It showed even lower sensitivity for regional metastasis detection in both per-patient and per-lesion analyses when restricted to patients referred for initial staging or those who underwent histopathology assessment. The lower sensitivities indicate that the negative [^18^F]F-FDG PET results in diagnosing regional lymph node metastasis were not highly accurate and may not be relied upon solely to rule out malignant involvement (high false negative results). Therefore, additional diagnostic procedures such as sentinel lymph node biopsy (SLNB) are warranted. These results are in accordance with earlier reviews, which have also emphasized the limited sensitivity of [^18^F]F-FDG PET imaging for detecting regional metastasis in melanoma patients [112,113,114,115,116]. In a meta-analysis conducted by Xing et al. [113], [^18^F]F-FDG PET showed 30% sensitivity in detecting regional nodal metastasis, significantly inferior to ultrasonography (US), with 60% sensitivity. However, it seems that even US cannot serve as a reliable tool in low-staged patients. Stahile et al. [90] investigated the value of US and [^18^F]F-FDG PET/CT in stage IIB/C melanoma patients prior to SLNB and reported that preoperative negative imaging could not guarantee the absence of sentinel lymph node metastases. Thus, SLNB could not be omitted from patient management, though US performed better than [^18^F]F-FDG PET. Similarly, in nearly all the studies included in the subgroup analysis for regional metastasis detection in our meta-analysis, the comparison of [^18^F]F-FDG PET and SLNB clearly indicated that [^18^F]F-FDG PET could not serve as a suitable substitute for SLNB. Notably, Schaarschmidt et al. [87] showed that [^18^F]F-FDG PET/MRI could not also replace SLNB since it did not reliably differentiate benign from malignant lymph nodes. Even with the addition of advanced sequences, including diffusion-weighted imaging (DWI), the sensitivity of [^18^F]F-FDG PET/MRI was not improved significantly.

Similarly, although this drawback is size-dependent, being more prominent in subcentimetric nodes, especially < 5 mm [34,37,115], in cases of thick primary melanomas the use of [^18^F]F-FDG PET/CT proved insufficient to replace the necessity of SLNB. Maubec et al. [117] only included patients with thick (>4 mm) primary melanoma in their study and reported that, even in these patients, SLNB remained the technique of choice for the most accurate regional staging. In higher stages of melanoma, however, [^18^F]F-FDG PET imaging can be a valuable tool before SLNB in cases of distant metastasis, up-staging patients regardless of their nodal status. In this regard, Hardie et al. [112] reported that conducting staging [^18^F]F-FDG PET/CT before SLNB in patients with pT4b melanoma could detect metastases in more than 20% of patients, changing the management plan without any treatment delay. However, they again recommended that SLNB should remain the crucial modality for achieving *definitive* staging.

Considering distant metastasis, [^18^F]F-FDG PET demonstrated reliable performance in both per-patient and per-lesion analyses, as well as in both initial and secondary staging groups. At the patient level, it had a pooled sensitivity and specificity of 88% and 94%, respectively. In the lesion-based analysis, these measures were 84% and 94%. These accurate results in [^18^F]F-FDG PET, a whole-body one-stop-shop imaging, make it a reliable modality for assessing patients, especially when there is a high suspicion of distant involvement (e.g., higher stages, unfavorable characteristics). However, it is important to acknowledge the certain limitations of [^18^F]F-FDG PET imaging in this regard. Due to the significant [^18^F]F-FDG uptake in normal brain tissue, [^18^F]F-FDG PET imaging exhibits low sensitivity in detecting brain metastases. For this purpose, MRI imaging is the preferred modality and should be conducted whenever the evaluation of the brain in melanoma patients is considered clinically relevant [79,118]. Although we could not assess this issue, [^18^F]F-FDG PET/MRI may hypothetically serve even better in detecting distant metastases, i.e., offsetting the limitation of [^18^F]F-FDG PET in the brain. Among the studies we evaluated, Berzaczy et al. [29] compared whole-body [^18^F]FDG-PET/MRI with [^18^F]FDG-PET/CT and reported that both modalities have comparable accuracy in detecting metastatic disease, though [^18^F]F-FDG PET/MRI was superior at detecting cerebral metastases. Therefore, when planning surgery or focused radiotherapy for patients with known or suspected melanoma brain metastases, [^18^F]F-FDG PET/MRI may be the preferred option. Replacing CT with MRI may also help detect equivocal/non-avid hepatic metastases. This is particularly true for small melanin-containing lesions (<8–10 mm) that are more effectively visualized with MRI due to their bright signal on T1-weighted imaging [49,79,119]. Having said that, if opting for [^18^F]F-FDG PET/MRI, performing additional diagnostic CT or close follow-up may still be needed because of the limitations in the pulmonary parenchyma.

It is also necessary to mention that in addition to the size of regional or distant metastatic lesions, the size of the primary lesion also plays a pivotal role in [^18^F]F-FDG PET diagnostic performance and, consequently, cost-effectiveness. Patients with a Breslow thickness of <1 mm have an exceedingly low risk for metastasis, rendering [^18^F]F-FDG PET not cost-effective [120,121]. Patients with an intermediate Breslow thickness (1–4 mm) face an elevated risk of locoregional metastasis but maintain a relatively low risk for distant metastasis (less than 20%). In this context, again, [^18^F]F-FDG PET may be of limited value, and SNLB remains the most accurate test for identifying locoregional spread [52,120,122]. Patients with Breslow thickness >4 mm have a higher risk of distant metastases; therefore, the use of imaging modalities, including [^18^F]F-FDG PET, seems beneficial in this group of patients.

[^18^F]F-FDG PET has also shown promising results in mucosal melanoma. Mucosal melanoma may sometimes exhibit distinct behavior compared to cutaneous melanoma, typically manifest at an advanced stage, displaying greater aggressiveness and leading to a poorer prognosis, regardless of the stage at which they are diagnosed [123]. However, as the data in this regard were sparse as it is a rare condition, pooled values could not be separately analyzed. Based on the limited literature we found on this subject, [^18^F]F-FDG PET may be of benefit in mucosal melanoma. In a study by Bakare et al. [22] investigating patients with anorectal melanoma, [^18^F]F-FDG PET/CT showed a sensitivity of 97%, specificity of 100%, and accuracy of 97% in detecting primary lesions in initial staging. Regional nodal involvement was correctly detected in 46/61 patients, with a false result (false negative) in only one case. Moreover, all distant metastases were detected. Considering suspected recurrence/restaging, [^18^F]F-FDG PET/CT showed an accuracy of 96%. Similarly, Agrawal et al. [14] evaluated the accuracy of [^18^F]F-FDG PET/CT in staging and restaging head and neck mucosal melanomas and reported accuracies of 100% for detecting primary lesions, 92% for regional nodal metastasis, and 86% for detecting distant metastasis.

Lastly, although there was scarce literature on the value of [^18^F]F-FDG PET in melanoma with unknown primary, and we could not perform exclusive analyses on these patients considering the diagnostic performance of [^18^F]F-FDG PET (e.g., group-specific staging, finding the primary site), there are some findings to mention. In the study by Eldon et al. [41], 12% of melanoma patients were of unknown primary origin, indicating a not uncommon type of patient that we may encounter in our daily routine in the clinic. Among them, 42% showed positive [^18^F]F-FDG PET and none of the patients with a negative scan showed active disease three months later. Interestingly, one third of patients underwent additional surgery because of the [^18^F]F-FDG PET findings, which may support its diagnostic value in this special group of patients. Other studies have also reported that up to 17% of melanoma patients may have an unknown primary origin [124]. In a study by Tos et al., they investigated only patients with unknown primary origin [125]. More than half of patients with cutaneous metastases underwent [^18^F]F-FDG PET. Again, in one third of patients, [^18^F]F-FDG PET could detect new metastases which were missed on other examinations, while having only one false positive finding. Similarly, in patients referred for lymph node metastases from unknown primary, approximately one third showed additional metastases. Thus, they could recommend that [^18^F]F-FDG PET be a step in staging melanoma from unknown origin, although it was not an efficient modality for finding the primary site. Likewise, in the study by Kole et al., [^18^F]F-FDG PET could not find the primary site in the majority of patients with melanoma, mainly because of the small size of the lesions, being below the spatial resolution threshold of [^18^F]F-FDG PET [126]. This finding was supported by Pelosi et al., who found none of the primary sites in four studied patients with melanoma [127]. Thus, overall, it seems that although [^18^F]F-FDG PET is not a satisfactory modality for finding an occult primary site in melanoma, its diagnostic accuracy in patient staging and metastasis detection is not affected in this specific population.

This study had limitations. Being aware of these limitations, readers can better interpret the findings and consider the potential impact of these factors on the study’s conclusions. The major problem was the observed considerable heterogeneity in the overall analysis. This heterogeneity could be attributed to several factors. First, in the overall analysis, we included studies evaluating the performance of [^18^F]F-FDG PET in different clinical instances, such as initial staging, restaging, regional detection, and distant metastasis. As discussed, [^18^F]F-FDG PET performs differently in each of these clinical scenarios, leading to the observed heterogeneity. By dividing studies into subgroups and conducting separate analyses, reduced heterogeneity was observed. Second, our study selection did not have a time limit, and the included articles were published from 1995 to 2021, covering the results of approximately three decades. This considerable time span might have contributed to the observed heterogeneity between studies due to the increasing experience in utilizing and interpreting [^18^F]F-FDG PET scans in addition to standardization of the protocols in cancer patients and advancements in imaging technology, leading to improved resolution of PET scans over the years. Third, a concern highlighted in this meta-analysis was the variability in methodologies and reference standards utilized across different studies. The inconsistent inclusion of a “gold standard” in these studies might lead to potential verification bias or results heterogeneity. We tried to mitigate this concern to some extent by including reliable reference standards and also limiting the results to histopathology evaluation as the gold standard when possible. Last, in our analysis, we incorporated all studies on non-ophthalmic melanoma, including cutaneous and mucosal melanoma, as well as melanoma of unknown origin. For example, as discussed, mucosal melanoma may behave differently compared to cutaneous melanoma. However, we excluded ophthalmic melanoma patients because of the consensus about the different nature of the disease.

## 5. Conclusions

In conclusion, our comprehensive meta-analysis provided compelling evidence supporting the importance of [^18^F]F-FDG PET imaging as a valuable, non-invasive diagnostic modality for non-ophthalmic malignant melanoma. [^18^F]F-FDG PET imaging demonstrated reliable performance in various clinical scenarios in both initial and secondary staging. However, our results indicated that despite its high specificity, it is less effective in detecting regional lymph node metastasis and could not replace SLNB. Nevertheless, [^18^F]F-FDG PET remains a critical imaging modality for guiding clinical decisions and optimizing patient care in melanoma management.

## Figures and Tables

**Figure 1 cancers-16-00215-f001:**
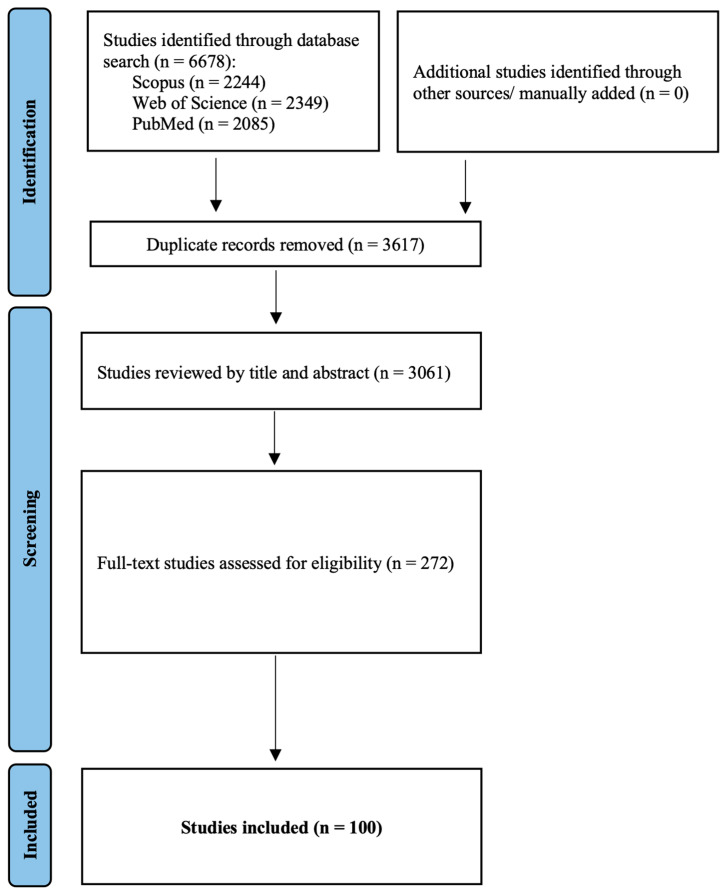
Study selection process flowchart.

**Figure 2 cancers-16-00215-f002:**
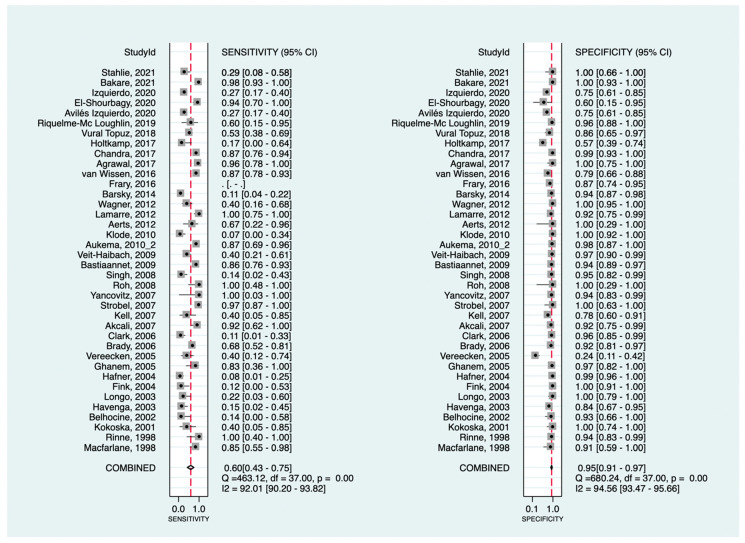
Forest plots for the pooled sensitivity and specificity calculation for initial staging at the patient level. Horizontal lines represent 95% confidence intervals of the individual studies [13,14,15,20,21,22,24,27,28,33,35,36,40,44,46,49,52,53,58,62,63,64,67,73,74,84,85,86,89,90,95,99,100,102,103,107,109].

**Figure 3 cancers-16-00215-f003:**
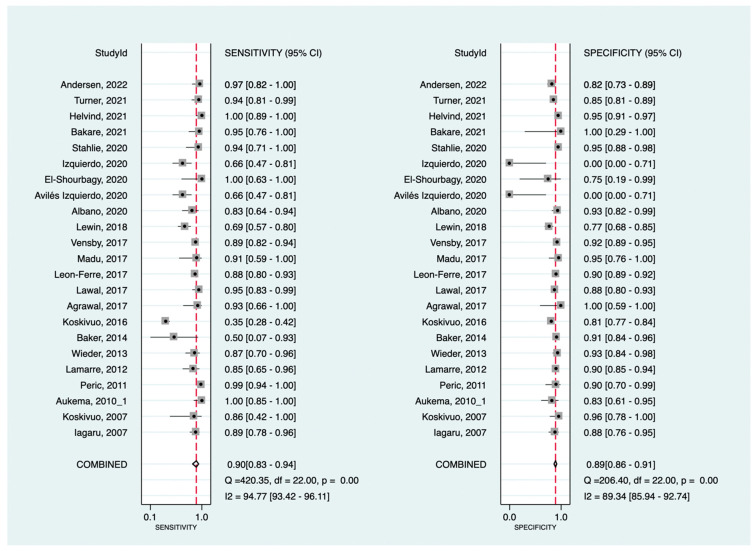
Forest plots for the pooled sensitivity and specificity calculation for secondary staging at the patient level. Horizontal lines represent 95% confidence intervals of the individual studies [14,16,17,19,21,22,23,40,54,60,65,66,67,69,71,72,75,78,91,97,101,108].

**Figure 4 cancers-16-00215-f004:**
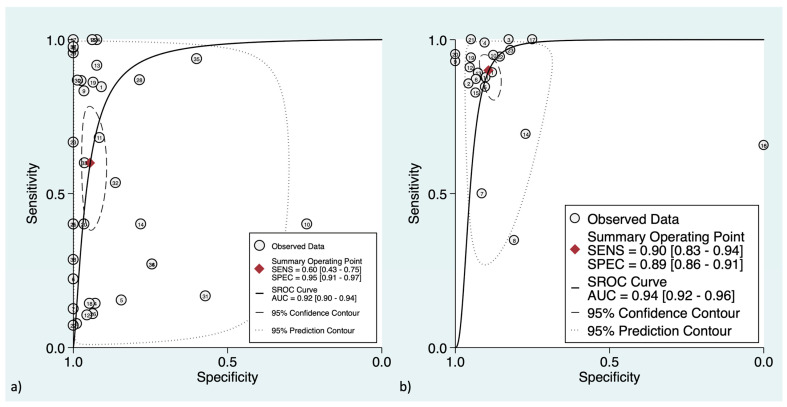
Hierarchical summary receiver operating characteristic curve for the patient-level assessment at (**a**) initial staging and (**b**) secondary staging. The “Observed Data” points show accuracy for each study, and the “Summary Operating Point” represents the pooled accuracy.

**Figure 5 cancers-16-00215-f005:**
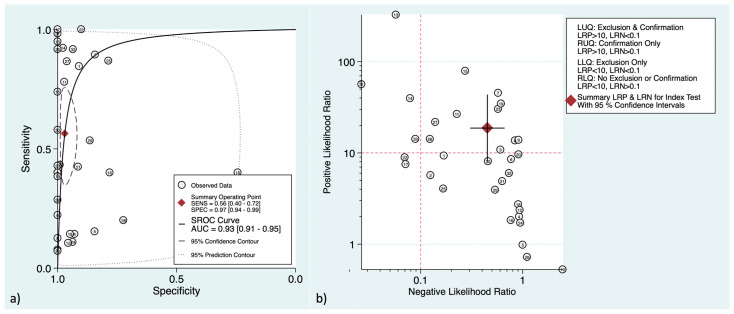
(**a**) Hierarchical summary receiver operating characteristic curve for the regional lymph node metastasis detection. The “Observed Data” points show accuracy for each study, and the “Summary Operating Point” represents the pooled accuracy. (**b**) The likelihood ratio scattergram for the regional lymph node metastasis diagnosis. AUC = area under the curve, SENS = sensitivity, SPEC = specificity, LUQ = left upper quadrant, RUQ = right upper quadrant, LLQ = left lower quadrant, RLQ = right lower quadrant, LRP = positive likelihood ratio, LRN = negative likelihood ratio.

**Figure 6 cancers-16-00215-f006:**
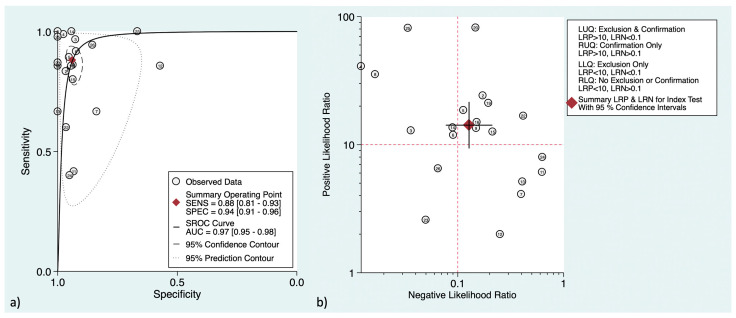
(**a**) Hierarchical summary receiver operating characteristic curve for the distant metastasis detection. The “Observed Data” points show accuracy for each study, and the “Summary Operating Point” represents the pooled accuracy. (**b**) The likelihood ratio scattergram for the distant metastasis diagnosis. AUC = area under the curve, SENS = sensitivity, SPEC = specificity, LUQ = left upper quadrant, RUQ = right upper quadrant, LLQ = left lower quadrant, RLQ = right lower quadrant, LRP = positive likelihood ratio, LRN = negative likelihood ratio.

**Figure 7 cancers-16-00215-f007:**
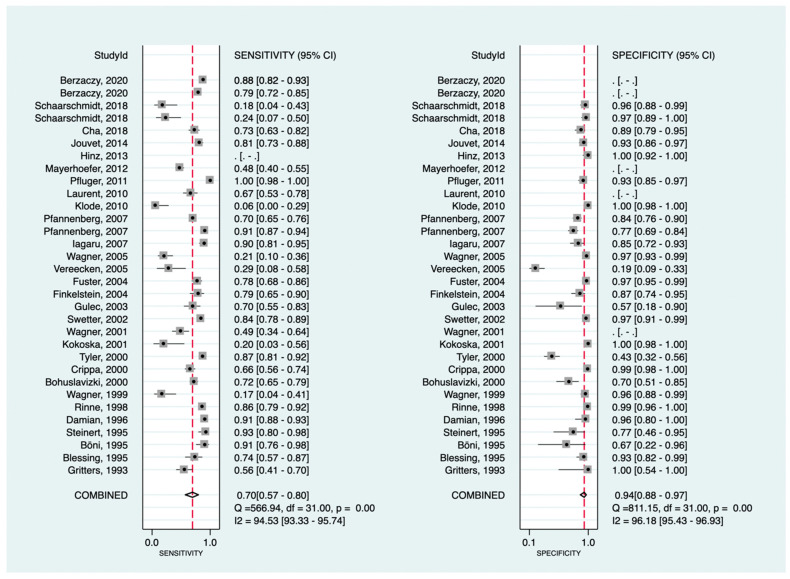
Forest plots for the pooled sensitivity and specificity calculation at the lesion level. Horizontal lines represent 95% confidence intervals of the individual studies [29,30,31,32,34,37,38,45,47,50,51,55,60,61,63,64,68,76,79,80,84,87,92,96,98,102,104,105,106].

**Table 1 cancers-16-00215-t001:** Characteristics of Included Studies.

Author, Year [Ref.]	Modality	Reference Standard	Level of Analysis	Number of Included Patients or Lesions
Acland, 2000 [12]	PET	Histopathology, Follow-up	P	52
Aerts, 2012 [13]	PET/CT	Capsule Endoscopy	P	9
Agrawal, 2017 [14]	PET/CT	Histopathology, Follow-up	P	19
Akcali, 2007 [15]	PET/CT	Histopathology	P	38
Albano, 2020 [16]	PET/CT	Histopathology, Follow-up	P	74
Andersen, 2022 [17]	PET/CT	Histopathology, MRI, Follow-up	P	124
Arrangoiz, 2012 [18]	PET/CT	Histopathology	P	56
Aukema, 2010_1 [19]	PET/CT	Histopathology, Further imaging, Follow-up	P	46
Aukema, 2010_2 [20]	PET/CT	Histopathology, Further imaging, Follow-up	P	70
Avilés Izquierdo, 2020 [21]	PET/CT	Histopathology	P	83
Bakare, 2021 [22]	PET/CT	Histopathology, Follow-up	P	61
Baker, 2014 [23]	PET/CT	Histopathology, Follow-up	P	38
Barsky, 2014 [24]	PET/CT	Histopathology	P	149
Bastiaannet, 2006 [26]	PET	Histopathology, Follow-up	P	257
Bastiaannet, 2009 [27]	PET	Histopathology, Further imaging, Follow-up	P	251
Bastiaannet, 2012 [25]	PET	Histopathology, CT scan, Follow-up	P	253
Belhocine, 2002 [28]	PET	Histopathology	P	21
Berzaczy, 2020 [29]	PET/CT	Histopathology, Follow-up	L	160
	PET/MRI			
Blessing, 1995 [30]	PET	Histopathology, Further imaging, Follow-up	L	83
Bohuslavizki, 2000 [31]	PET	Histopathology	L	189
Böni, 1995 [32]	PET	Histopathology, Further imaging	L	39
Brady, 2006 [33]	PET	Histopathology, CT scan, Follow-up	P	103
Cha, 2018 [34]	PET/CT	Histopathology	L	165
Chandra, 2017 [35]	PET/CT	Histopathology, Follow-up	P	70
Clark, 2006 [36]	PET	Histopathology	P	64
Crippa, 2000 [37]	PET	Histopathology	P	38
			L	56
Damian, 1996 [38]	PET	Histopathology, Further imaging	L	639
Eigtved, 2000 [39]	PET	Histopathology, Further imaging, Follow-up	P	38
El-Shourbagy, 2020 [40]	PET/CT	Histopathology, CT scan	P	50
Eldon, 2017 [41]	PET/CT	Histopathology, Further imaging, Follow-up	P	58
Essler, 2011 [42]	PET/CT	Histopathology, Further imaging, Follow-up	P	125
Falk, 2007 [43]	PET/CT	Histopathology, Further imaging, Follow-up	P	60
Fink, 2004 [44]	PET/CT	Histopathology	P	48
Finkelstein, 2004 [45]	PET/CT	Histopathology, Follow-up	L	94
Frary, 2016 [46]	PET/CT	Histopathology, Follow-up	P	46
Fuster, 2004 [47]	PET	Histopathology, CT scan, Follow-up	L	146
Gellén, 2015 [48]	PET/CT	Histopathology, Follow-up	P	97
Ghanem, 2005 [49]	PET	MRI, Follow-up	P	35
Gritters, 1993 [50]	PET	Histopathology, Further imaging, Follow-up	P	12
			L	52
Gulec, 2003 [51]	PET	Histopathology, Further imaging	L	44
Hafner, 2004 [52]	PET	Histopathology, Follow-up	P	100
Havenga, 2003 [53]	PET	Histopathology	P	45
Helvind, 2021 [54]	PET/CT	Histopathology	P	138
Hinz, 2013 [55]	PET/CT	Histopathology	L	59
Holder, 1998 [56]	PET	Histopathology, CT scan	P	76
Holtkamp, 2017 [58]	PET/CT	Histopathology, Follow-up	P	41
Holtkamp, 2020 [57]	PET/CT	Histopathology, Follow-up	P	25
Horn, 2006 [59]	PET	Histopathology, Further imaging, Follow-up	P	33
Iagaru, 2007 [60]	PET/CT	Histopathology, Follow-up	P	106
			L	139
Izquierdo, 2020 [21]	PET/CT	Histopathology	P	83
Jouvet, 2014 [61]	PET/CT	Histopathology, Follow-up	L	191
Kell, 2007 [62]	PET/CT	Histopathology	P	37
Klode, 2010 [63]	PET/CT	Histopathology	P	61
			L	174
Kokoska, 2001 [64]	PET	Histopathology	P	18
			L	63
Koskivuo, 2007 [66]	PET	Histopathology, Follow-up	P	30
Koskivuo, 2016 [65]	PET/CT	Histopathology, Follow-up	P	110
Lamarre, 2012 [67]	PET/CT	Histopathology, Follow-up	P	19
Laurent, 2010 [68]	PET/CT	Histopathology, Further imaging, Follow-up	L	120
Lawal, 2017 [69]	PET/CT	Histopathology, Follow-up	P	144
Lazaga, 2013 [70]	PET/CT	Histopathology, Follow-up	P	200
Leon-Ferre, 2017 [71]	PET/CT	Histopathology, Follow-up	P	299
Lewin, 2018 [72]	PET/CT	Histopathology, Further imaging, Follow-up	P	47
Longo, 2003 [73]	PET	Histopathology	P	25
Macfarlane, 1998 [74]	PET	Histopathology	P	22
Madu, 2017 [75]	PET/CT	Histopathology, Further imaging, Follow-up	P	18
Mayerhoefer, 2012 [76]	PET/CT	Histopathology, Follow-up	P	38
			L	181
Paquet, 2000 [77]	PET	Histopathology, Follow-up	P	24
Peric, 2011 [78]	PET/CT	Histopathology, Further imaging	P	115
Pfannenberg, 2007 [79]	PET/CT	Histopathology, Follow-up	L	420
	PET			
Pfluger, 2011 [80]	PET/CT	Histopathology, Follow-up	L	232
Prakoso, 2011 [81]	PET/CT	Capsule Endoscopy, Follow-up	P	21
Querellou, 2010 [82]	PET/CT	Histopathology, Follow-up	P	189
Reinhardt, 2006 [83]	PET/CT	Histopathology, Follow-up	P	250
	PET			
Rinne, 1998 [84]	PET	Histopathology, Further imaging, Follow-up	P	100
			L	121
Riquelme-Mc Loughlin, 2019 [85]	PET/CT	Histopathology, Further imaging, Follow-up	P	61
Roh, 2008 [86]	PET	Histopathology	P	10
Schaarschmidt, 2018 [87]	PET/MRI	Histopathology	L	87
	PET/CT			
Schauwecker, 2003 [88]	PET	Histopathology	P	119
Singh, 2008 [89]	PET/CT	Histopathology	P	54
Stahlie, 2020 [91]	PET/CT	Histopathology	P	35
Stahlie, 2021 [90]	PET/CT	Histopathology	P	23
Steinert, 1995 [92]	PET	Histopathology, Further imaging	L	53
Strobel, 2007 [94]	PET/CT	Histopathology, Further imaging, Follow-up	P	124
	PET			
Strobel, 2007 [95]	PET/CT	Histopathology, Further imaging, Follow-up	P	47
Strobel, 2009 [93]	PET/CT	Histopathology, Further imaging, Follow-up	P	14
Swetter, 2002 [96]	PET	Histopathology, Further imaging, Follow-up	L	199
Turner, 2021 [97]	PET/CT	Histopathology, Further imaging, Follow-up	P	332
Tyler, 2000 [98]	PET	Histopathology	L	234
van Wissen, 2016 [99]	PET/CT	Histopathology	P	70
Veit-Haibach, 2009 [100]	PET/CT	Histopathology, Further imaging, Follow-up	P	56
Vensby, 2017 [101]	PET/CT	Histopathology, Further imaging, Follow-up	P	238
Vereecken, 2005 [102]	PET/CT	Histopathology, Follow-up	P	43
			L	43
Vural Topuz, 2018 [103]	PET/CT	Histopathology	P	65
Wagner, 1999 [104]	PET	Histopathology	L	89
Wagner, 2001 [106]	PET	Histopathology	L	49
Wagner, 2005 [105]	PET	Histopathology, Follow-up	L	184
Wagner, 2012 [107]	PET/CT	Histopathology	P	49
Wieder, 2013 [108]	PET/CT	Histopathology, Follow-up	P	28
Yancovitz, 2007 [109]	PET/CT	Histopathology, Further imaging, Follow-up	P	36
Zimmermann, 2021 [110]	PET/CT	Histopathology, Follow-up	P	44

Note.—Ref. reference number; CT. computed tomography; MRI. magnetic resonance imaging; PET. positron emission tomography; P. patient; L. lesion.

## Data Availability

All datasets and analyses are available and can be accessed upon a reasonable request from the corresponding author.

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
