# Peer review of "Diagnostic Performance of [18F]F-FDG Positron Emission Tomography (PET) in Non-Ophthalmic Malignant Melanoma: A Systematic Review and Meta-Analysis of More Than 10,000 Melanoma Patients"

_cancers, 2024, doi:10.3390/cancers16010215_

Round 1
Reviewer 1 Report
Comments and Suggestions for Authors
Dear Authors,
you did a large effort to review all studies of [18F]F-FDG PET and evaluate its diagnostic performance in different melanoma clinical setting. Personally, I found interesting the low sensitivity of [18F]F-FDG PET for lymph-node involvement, as an isolated result compared to a general higher accuracy. In addition, in the future, I'd suggest to include more recently articles (i.e. last 10 years) to obtain more homogeneous results.
Some minor comments:
- Can you add a paragraph in discussion on the value of [18F]F-FDG PET for melanoma of unknown origin, although the sparse data in the literature?
- Except to lymph-nodes assessment, and according to your results, do you believe that this technique may be considered a valuable alternative to PET-TC in the follow-up of patients with melanoma? Please add a sentence in discussion.
Best Regards.
Author Response
Distinguished Reviewer #1:
Dear Authors,
you did a large effort to review all studies of [18F]F-FDG PET and evaluate its diagnostic performance in different melanoma clinical setting. Personally, I found interesting the low sensitivity of [18F]F-FDG PET for lymph-node involvement, as an isolated result compared to a general higher accuracy. In addition, in the future, I'd suggest to include more recently articles (i.e. last 10 years) to obtain more homogeneous results.
A: Many thanks for your comment. We entirely agree with your point of view and will take it into account in future investigations.
Some minor comments:
- Can you add a paragraph in discussion on the value of [18F]F-FDG PET for melanoma of unknown origin, although the sparse data in the literature?
A: Sure. We added a paragraph in this regard (before limitations).
- Except to lymph-nodes assessment, and according to your results, do you believe that this technique may be considered a valuable alternative to PET-TC in the follow-up of patients with melanoma? Please add a sentence in discussion.
A: We are afraid that we did not get your point. We believe there was a typo (PET-TC). However, we did not understand what technique could be an alternative to PET/CT. If you think we should implement this comment to enhance the discussion we would be more than glad to explain it more a bit.
Best Regards.
Reviewer 2 Report
Comments and Suggestions for Authors
This manuscript is a comprehensive, real world meta-analysis which provides evidence supporting the importance [ 18F]F-FDG PET imaging as a valuable, non-invasive diagnostic tool for non-ophthalmic malignant melanoma. The intensive and well designed analysis indicates that despite the high specificity of 18F]F-FDG PET imaging, it is less effective in detecting regional lymph node metastasis and could not replace showed low sensitivity in detecting regional lymph node metastasis and could not replace lymph node biopsy. Nevertheless, [ 18F]F-FDG PET remains a critical imaging modality for guiding clinical decisions and optimizing patient care in melanoma management.
Comment: The references should be formatted
The text on the Figures are too small
Author Response
Distinguished Reviewer #2:
This manuscript is a comprehensive, real world meta-analysis which provides evidence supporting the importance [ 18F]F-FDG PET imaging as a valuable, non-invasive diagnostic tool for non-ophthalmic malignant melanoma. The intensive and well designed analysis indicates that despite the high specificity of 18F]F-FDG PET imaging, it is less effective in detecting regional lymph node metastasis and could not replace showed low sensitivity in detecting regional lymph node metastasis and could not replace lymph node biopsy. Nevertheless, [ 18F]F-FDG PET remains a critical imaging modality for guiding clinical decisions and optimizing patient care in melanoma management.
Comment: The references should be formatted
A: Thanks for the point. We re-checked the references to follow the journal’s guidelines.
The text on the Figures are too small
A: Thanks for your comment. The figures (Figures 4-6) were replaced. I hope the new ones are considered satisfactory.
Reviewer 3 Report
Comments and Suggestions for Authors
Dear Editor,
I was pleased to read this interesting systematic review and meta-analysis on the diagnostic performance of [18F] F-FDG PET imaging for malignant melanoma. The manuscript is the largest in his field, is clearly structured and cites the most recent literature relevant to the topic. This is a current area of interest in malignant melanoma, thus, accurate staging and treatment monitoring are crucial for optimal management.
The conclusions are consistent with the purpose of this research, in order to provide an accurate imaging tool for diagnostic purpose in a highly agressive skin tumor.
There are some few errors in the references: 5,19, 27,35,37,50,65,87,117, 120-122
There are no specific improvements to be done.
Author Response
Distinguished Reviewer #3:
Dear Editor,
I was pleased to read this interesting systematic review and meta-analysis on the diagnostic performance of [18F] F-FDG PET imaging for malignant melanoma. The manuscript is the largest in his field, is clearly structured and cites the most recent literature relevant to the topic. This is a current area of interest in malignant melanoma, thus, accurate staging and treatment monitoring are crucial for optimal management.
The conclusions are consistent with the purpose of this research, in order to provide an accurate imaging tool for diagnostic purpose in a highly agressive skin tumor.
A: Thanks. That meant a lot. We are glad that we could present our points satisfactorily.
There are some few errors in the references: 5,19, 27,35,37,50,65,87,117, 120-122
A: Thanks for the point. However, we did not get the point of whether these errors are due to the mis-insertion of the references or they do not follow the journal’s guidelines.
There are no specific improvements to be done.